# Roles of a Y-Linked iDmrt1 Paralogue and Insulin-like Androgenic Gland Hormone in Sexual Development in the Tropical Rock Lobster, *Panulirus ornatus*

**DOI:** 10.3390/ijms26115149

**Published:** 2025-05-27

**Authors:** Ai Hang Tran Nguyen, Jihye Yoon, Wenyan Nong, Susan Glendinning, Quinn P. Fitzgibbon, Gregory G. Smith, Jerome H. L. Hui, Ka Hou Chu, Volker Herzig, Tomer Ventura

**Affiliations:** 1Centre for BioInnovation, University of the Sunshine Coast, Sippy Downs, QLD 4556, Australia; nat018@student.usc.edu.au (A.H.T.N.); sglendinning@usc.edu.au (S.G.);; 2School of Science, Technology and Engineering, University of the Sunshine Coast, Sippy Downs, QLD 4556, Australia; 3Institute for Marine and Antarctic Studies, University of Tasmania, Private Bag 49, Hobart, TAS 7001, Australia; 4School of Life Sciences, Simon F.S. Li Marine Science Laboratory, State Key Laboratory of Agrobiotechnology, Institute of Environment, Energy and Sustainability, The Chinese University of Hong Kong, Shatin, NT, Hong Kong SAR, China; 5Southern Marine Science and Engineering Guangdong Laboratory (Guangzhou), Guangzhou 511458, China; 6School of Life Sciences, The Chinese University of Hong Kong, Shatin, NT, Hong Kong SAR, China

**Keywords:** master sex determinant, Dmrt, iDMY, IAG, transactivation domain, Crustacea

## Abstract

Understanding the mechanisms of sexual development would pave the way for producing mono-sex populations to aid the aquaculture industry. This study investigates the functions of the Y-linked *iDmrt1* paralogue (*Po-iDMY)* and insulin-like androgenic gland hormone (Po-IAG) in the process of sexual development in the tropical rock lobster, *Panulirus ornatus* (TRL). Previously, we identified that *Po-iDMY*, a male-specific heterogametic (Y-linked) paralogue of the autosomal *Po-iDmrt1* found in TRL, is a second sex-linked *iDmrt* gene identified in invertebrates. Using 5′ and 3′ rapid amplification of cDNA ends and data from a draft male genome (with an assembly genome size of approximately 2.446 Gbp and 87% BUSCO completeness), we obtained the full-length *Po-iDMY* gene (encoding a protein of 312 amino acids). A 411 bp male-specific sequence located at the 3′ untranslated region of *Po-iDMY* mRNA was used as a sex marker, which was reported for the first time in our draft genome. However, *Po-iDMY* is not a master sex-determining factor since it was not expressed across developmental stages of embryos, juveniles and adults. Instead, we silenced *Po-IAG* at an early juvenile stage, generating two potential neo-females, implying that sexual manipulation could be a promising technique in TRL.

## 1. Introduction

Over 67,000 species comprise the subphylum Crustacea, which occupies a varied range of ecological niches and exhibits diverse life histories [1,2,3,4]. Decapods are the most diverse order of Crustacea and include commercially significant species of Penaeidae (shrimp) [5], Palaemonidae (prawns) [6], Parastacidae (crayfish) [7], Portunidae (crabs) [8], and Palinuridae (spiny lobsters) [9], etc. [1]. The value of global crustacean aquaculture exceeded USD 79.5 billion in 2020 [10]. Among them, the tropical rock lobster *Panulirus ornatus* (TRL) is among the most valuable seafoods in the world [11,12,13], and its demand is increasing in the global food market, especially in Hong Kong and mainland China [14,15,16,17]. This trend has driven research to optimize closed-life-cycle aquaculture to increase yield and profit and reduce the need to collect animals from the wild [18,19,20,21]. However, TRL aquaculture research has faced numerous challenges over the past decade. The most significant obstacle is the lengthy and complex larval phase, which can last for hundreds of days [22,23,24]. Maintaining healthy larvae during this prolonged period requires advanced aquaculture technologies and a high level of expertise, including the preparing of expensive natural foods such as mussels and live artemia [15,22,23,25]. Moreover, juvenile lobsters display considerable aggression and cannibalistic tendencies, emphasizing the urgent need for effective formulated feeds, which are currently unavailable [22,26,27]. As a result, recent TRL aquaculture research has focused on feeds and feeding [28,29,30,31,32], environmental conditions [33,34,35,36,37,38,39], and cannibalism [22,27,31,40,41,42,43]; however, little effort has been concentrated on creating mono-sex populations of TRL [44]. In TRL, during the breeding season, the growth rate decreases in both males and females after their first mating. Females allocate 42% of their total weight gain to egg production [45,46] and experience severe mortality rates that have not been reported in other lobster species [47,48,49]. Therefore, sexual manipulation to attain a mono-sex culture could reduce weight loss and mortality during the mating season. Further insights into the regulation of the sexual development pathway of TRL are therefore required for developing mono-sex stocks [50].

Genomic and transcriptomic resources provide essential tools for studying fundamental biological processes, including reproduction and sexual development [51,52,53,54,55]. In the last decade, transcriptomic libraries have revealed a significant number of genes with sexually dimorphic expression across various animal species. Among them, the Dsx- and mab-3-related transcription factor (*Dmrt*) gene family is extensively well described in sex determination and differentiation pathways across the animal kingdom [56,57,58,59]. *Dmrt* genes were discovered through a comparison of the functional and molecular structures of doublesex (*dsx*) from the fruitfly *Drosophila melanogaster* with male abnormal 3 (mab-3) from the nematode worm *Caenorhabditis elegans* [60,61,62]. *Dmrt* genes encode a zinc finger motif called the DNA-binding domain sequence, known as the DM domain [56,60,63]. Recent evidence shows that the *Dmrt1* gene (*doublesex and mab-3 related transcription factor 1* gene) is vital to male sexual development in various vertebrates (from fish to mammals) and in invertebrates (e.g., flies and worms) for regulating the process of gonadal development and sexual differentiation [59,64,65,66]. Notably, even in crustaceans, where the sex determination system is remarkably complex and diverse in theory and practice [67,68,69], *iDmrt1* emerges as a pivotal upstream factor in the integrated sexual development cascade [57,70]. According to Chandler 2017 [62], the first heterogametic sex-linked *iDmrt1* gene was discovered in the male genome of the eastern spiny (rock) lobster (ERL), *Sagmariasus verreauxi*, namely, *Sv-iDMY*. In this work, the authors found that *Sv-iDMY* was a male-specific sequence that was used as a genetic sex marker [62]. Additionally, Sv-iDMY was found to be expressed at an early stage in embryonic development that dominantly suppressed its autosomal paralogue (*iDmrt1*), indicating that Sv-iDMY is a master sex regulator in ERL. Structurally, *Sv-iDMY* is a Y-linked paralogue of an autosomal *iDmrt1* gene, and the protein shares conservation at the N terminus (N′)—DM domains of the protein coded by their somatic gene; however, there was a truncation in the sequence resulting in loss of the C terminus (C′). As a result, while Sv-iDMY and Sv-iDmrt1 share the conserved N′, which characterizes the *Dmrt* family, the non-conserved and truncated C′ tail demonstrates the flexibility of this protein in regulating the transcriptional processes as well as the sex determination pathway [62]. Indeed, transcriptional activation needs two transcriptional activators: (1) a DNA—binding domain (DM-domain) localizing the specific DNA sequences within genomic DNA located at the N’, and (2) a transactivation domain (TAD) dictating the extent and timing of up-regulation of a targeted gene located at the C’ [71,72]. As a result, the C′ function acts as a transcriptional activator through the TAD in its sequence. In the case of *S. verreauxi*, Sv-iDMY was unable to activate transcription and acted as a negative suppressor of Sv-iDmrt1 when lacking the TAD motif and a truncated C’ in comparison with their autosomal Sv-iDmrt. Moreover, the C’ of the Dmrt family has a second crucial role in maintaining binding stability and ensuring successful activation during the transcriptional process. Mechanistically, the C′ terminal recognition helix inserts into the DNA central groove while the DM-domain binds to the minor groove, resulting in stabilization of the DNA–protein junction and assisting in the assembly of the Dmrt—binding complex [73,74,75,76,77]. Ventura et al. (2020) have recently identified a second sex-linked iDmrt, *Po-iDMY*, and its autosomal *Po-iDmrt* by using a multi-tissue transcriptomic library of TRL tissues [78]. Interestingly, the central role of Po-iDMY in sexual determination may have been maintained in TRL when the Po-iDMY protein sequence was conserved between two spiny lobsters (ERL and TRL), suggesting *Po-iDMY* as a potential genetic sex marker and master sex determinant in these species. Notwithstanding, the expression at the early stage of embryo development and the protein architectures (the DM-domain at the N’ and the transactivation domain at the C’) of *iDMY* and *iDmrt* needs to be investigated to confirm their function in TRL.

In addition, the insulin-like androgenic gland hormone (IAG) is another crucial downstream mediator governing sexual differentiation [79], affecting the appearance of primary and secondary male sexual characteristics [80,81]. Since its first discovery in decapods in 2007, within two decades, IAG was discovered in many additional decapods [82,83,84,85,86,87,88,89]. Despite this, silencing *IAG* in decapods does not always result in successful sex change, as IAG is not a sex-determining factor [88,90,91,92,93]. In the case of *S. verreauxi*, the investigation initially focused on the androgenic gland developing at later life stages, well into maturity, meaning that gonad differentiation could not be entirely influenced by IAG in this species, as was the case in *Macrobrachium rosenbergii* [94,95,96]. Hence, it then became clear that the sexual development pathway in rock lobster differs from that established in the freshwater—prawn. The study on ERL suggested that other factors may be involved in regulating male sexual differentiation in spiny lobsters [97,98]. Previous research confirmed that *Sv-iDMY* plays a role as the master sexual development regulator upstream of *IAG* [62], with the potential to act as a target for inducing sex change by RNA interference in ERL. Thus, to better understand the regulation of the sexual development pathway in TRL, it seems prudent to identify Dmrt orthologues as a starting point since they have such an important role to play in the closely related ERL.

To date, two studies on the genome of TRL have been published: a female genome assembly, which is incomplete and fragmented [99], and a chromosome-level male genome [100]. Here, we describe the assembly of a second male genome for TRL, which was required to aid in identification of iDMY orthologue sequences in TRL. Our significant improvement in this draft male genome involves using our extensive transcriptomic data from various developmental stages and multiple tissues for TRL [78,101,102], encompassing sequences from nineteen different tissues, including reproductive tissues, therefore increasing the scope of the TRL genome to involve the annotation of genes related to sexual development and sex determination. We used our extensive transcriptomic data to aid in scaffolding the genome assemblies to recover gene regions or complete transcribed regions, significantly improving the draft genome quality and gene model length [103,104,105].

In the current study, we assembled the TRL male genome by combining Illumina NovaSeq 6000 (10 X linked-read files) with Pacific Biosciences (PacBio) sequencing. Then, we applied L_RNA_scaffolder to produce genome scaffolds, annotate protein-coding genes, and identify genes for functional research [103]. Using the male genome, we determined the full sequence of *Po-iDMY*. Since *Po-iDMY* expression is negligible across embryos, developmental metamorphic stages, and adult tissues according to our findings, it is likely that this gene is not the master sex regulator in TRL, although it is useful as a genetic sex marker in TRL. Moreso, being tightly linked with genetic sex, investigation of the newly established male genomes in conjunction with the extensive transcriptomic data can focus on regions at the *Po-iDMY* vicinity to identify other genes that are likely to serve as the master sex-regulating genes in TRL. The timing of *Po-IAG* expression prior to the development of sexual characteristics in TRL highlights its importance in TRL sexual differentiation. Finally, sex-reversal trials through *Po-IAG* gene silencing (using *Po-iDMY* as a genetic sex marker) suggest that this is a promising technology for producing TRL mono-sex populations in an aquaculture setting.

## 2. Results

### 2.1. Panulirus Ornatus Male Genome

Using the partial protein-coding sequence of *Po-iDMY* (191 aa) and the full sequence of IAG (165 aa) from our previous study [78], we searched an already published genome [100]. However, we could not find full-length sequences for those genes. The query coverages were 43% for *Po-iDMY* and 79% for *Po-IAG* (Appendix A). Therefore, we produced a draft male genome for TRL with support from highly detailed RNA-Seq data, including reproductive-related tissues such as the testis, ovary, oviduct, and the proximal, medial, and distal sperm duct regions from a previously reported research by Ventura et al. (2020) [78].

#### 2.1.1. Genome Survey

Adult TRL male (Figure 1A) genome characteristics were surveyed using GenomeScope [106] (k = 21) to estimate the size, heterozygosity, and percentage of repetitive elements present. A maximum depth from distribution data based on k-mer calculated the estimation of the genome size of 2.46 giga base pairs (Gbp) (Figure 1B,C), which was smaller than the results reported by [99] on the estimated female genome (3.23 Gbp). The heterozygosity and expected repetitive contents (REs) of the genome were estimated at 1.45% (at kcov = 19.3) and 46.8%, respectively (Figure 1C).

#### 2.1.2. Genome Sequencing Assembly

The TRL male genome was assembled using Chromium 10× genomic libraries (Illumina) data (1,473,906,560 read pairs) scaffolded with the Pacific BioSciences library (PacBio) (7,620,300 read pairs) and RNA-Seq data (see Appendix A). The sequencing results are available in the NCBI projects PRJNA952321, PRJNA952321, and PRJNA903480. The results showed that the assembled genome size of male TRL was 2.446 Gbp, containing 201,695 scaffolds with an N50 of 76 kbp and 5.24% of unknown nucleotides (N-count) (see Figure 1B,C). Total BUSCOs completeness analysis resulted in the detection of approximately 87% signature arthropod_odb10 (with 79.3% complete and single-copy BUSCOs (S) and 7.7% complete and duplicated BUSCOs (D)). In addition, the percentage of fragmented BUSCOs (F) and missing BUSCOs (M) in our genome were 8.8% and 4.2%, respectively (Figure 1C). The assembled male and female genomes and the annotated genes and predicted proteins can be found in Figshare [107].

#### 2.1.3. Repeat Annotation

Earl Grey pipeline v.4.0.5 [108] was run with RepeatMasker v.4.1.4 and RepeatModeler2 v.2.0.4 de novo to detect and annotate the repetitive contents in the assembled male genome. More than a quarter (26.71%) of the male genome was annotated as repetitive sequences (REs) (Figure 1D). About 16.11% of the genome was labeled as unclassified repeats, a small proportion was predicted to be simple repeats, microsatellites, and RNA (0.12%), and another 10.48% was estimated to be transposable elements (TEs). Of the TE portion of the TRL male genome, the majority of the annotated TEs were LINEs (5.33%) and DNA elements (2.72%), with only a small attribution from SINEs (0.12%) and LTRs (0.87%) (Figure 1D).

#### 2.1.4. Gene Prediction

We conducted gene prediction using the BRAKER3 pipeline and the Arthropoda proteins from OrthoBD11. The results revealed 129,814 protein-coding genes, with an average length of 214 aa for the TRL male genome. In an effort to evaluate the quality of this annotation, protein sets were analyzed with BUSCOs using the arthropod_odb10 lineage. These results unveiled 88.9% coverage of the total BUSCOs, with 53.5% coverage of complete and single-copy and 35.4% coverage of complete and duplicated of the 1013 examined genes (Figure 1C).

#### 2.1.5. Identification and Validation of the Male Y-Linked iDmrt Paralogue, Po-iDMY

The partial *Po-iDMY* transcript sequence was identified in a previous study by using a multi-tissue transcriptomic library of TRL [78]. Based on this finding, primers were designed to obtain the complete transcript of *Po-iDMY* using RACE and then validated by Sanger sequencing. We successfully elongated the 3′ end corresponding to the C’ tail of the Po-iDMY putative protein and obtained a male-specific sequence (411 bp; see Appendix A). Notably, we could not locate the male-specific sequence when we blasted it against the recently published chromosome-level genome [100]. However, the 5′ region, corresponding to the N’, could not be elongated using the 5′ RACE. As an alternative, we used the data from our annotated male genome to identify the complete Po-iDMY protein-encoding sequence (see Appendix A). As a result, a 939 bp ORF region encoding 312 amino acids (aa) was annotated and named the transcript *Po-iDMY* (PP530243). The theoretical isoelectric point and the molecular weight of the *Po-iDMY* protein were 8.63 and 34.59 k_D_, respectively. Using NCBI conserved domain tool version v3.20 to predict the conserved domain architecture, the protein had two DM structural domains located at 65–115 aa and 134–187 aa with a TAD (at 194–202 aa) in the sequence. To our knowledge, *Po-iDMY* is the second sex-linked *iDmrt* gene on the Y chromosome that has been identified in an invertebrate species. In addition, *Po-iDmrt1* (PP536547), *Po-iDMY* ’s autosomal paralogue, was previously identified in TRL [78]. In the present study, the predicted *Po-iDmrt1* was found to encode 510 aa and contained two DM domains located at 17–68 aa and 106–159 aa, respectively, and a TAD located at 166–174 aa. The putative Po-iDmrt1 and Po-iDMY proteins both contained two N’ DM domains and a TAD at the C’. However, Po-iDMY had a truncated C’ tail (312 aa) compared with the complete Po-iDmrt1 sequence (510 aa).

To validate the use of the *Po-iDMY* sequence as a genetic sex marker, forty pleopod samples from adult TRLs were used to extract genomic DNA in a blind trial. Twenty animals of each sex were used for PCR amplification of the short male-specific *Po-iDMY* sequence. A short male-specific band (208 bp) was amplified for the male lobsters, which was absent in the females (Figure 2). As a result, we concluded that the sex-specific marker could be used to correctly identify 21 male and 19 female samples. However, one sample failed; hence, the identification accuracy of this sex marker was nearly 100%. *Po-DMRT* was used as a positive control for the female samples to ensure DNA integrity (Appendix A). All samples except sample 21 had a band at 600 bp for the positive control. However, this sample was successfully used to amplify the *Po-iDMY* gene (see Figure 2), indicating the intact DNA of this sample.

### 2.2. iDMY and iDmrt Ontogeny

Broad spatial expression profiles for Po-iDMY and Po-iDmrt were generated from previously published transcriptomes of 11 stages of embryogenesis, 12 stages of larval development, and 18 adult tissues (a total of 121 RNA-Seq fastq files). A Basic Local Alignment Search Tool (BLAST version 2.16.0+) search across the CrustyBase database determined the FPKM values of *Po-iDMY* transcripts and plotted them for the three databases. According to the results, the expressions of *Po-iDMY* and *Po-iDmrt* were insignificant across embryogenesis, metamorphic stages, and the adult tissues (Appendix A). It is crucial to note that the *Po-iDMY* expression during embryogenesis was investigated to assess whether *Po-iDMY* has potentially evolved as a master sex-determining factor in TRL. In this study, the amino acid sequence coding for the DM domain (in the *Dmrt* family of genes) published on CrustyBase (https://crustybase.org/ (accessed on 24 February 2025)) [109], which functions as a transcription factor in the protein, was calculated as reads per kilobase per million reads (RPKM) in RNA-Seq libraries of multiple TRL embryonic stages (Figure 3). Interestingly, although several genes in the *Dmrt* family were expressed in male embryos (*99B*, *11E*, *DSX*), none of these factors were male-specific [78]. Our findings indicated no evidence supporting the notion that the male-specific gene *Po-iDMY* has a master sex-determining function in this species.

### 2.3. IAG Expression and the Appearance of Sexual Characteristics

We have shown that in TRL, *iDMY* is not likely to play a key role in sex determination. Another factor involved in developing male sexual characteristics in other species, Po-IAG, may be important in governing male sexual differentiation in TRL by regulating the development of male sexual characteristics in *P. ornatus*. In the context of this work, it is of note that the first external evidence of primary sexual differentiation appears in J4, seen in the emergence of the gonopores at the base of each of the third walking legs in females and fifth walking legs in males. RT-PCR was performed on RNA extracted from tissues at the fifth walking legs (including the androgenic gland area) to investigate the temporal expression pattern of *Po-IAG* in J1. The findings revealed that *Po-IAG* was expressed at this stage (J1) (Figure 4) before the development of sexual characteristics such as the gonopores. In addition, using our genetic sex marker, *Po-iDMY*, to identify male and female juveniles, we identified that *Po-IAG* was expressed in both males and females, which indicates that *Po-IAG* is not male-specific in TRL (Figure 4).

### 2.4. Po-IAG Silencing in Juvenile Panulirus Ornatus: Effects on Sexual Development

A trial was conducted to investigate a potential application of *Po-IAG* silencing to induce sex reversal from male to female lobsters, resulting in only 10 out of 66 animals surviving. It is worth noting that there was a high mortality rate of 63% after two injections. Seven of the surviving animals were phenotypically males, while three appeared female, as identified by visually examining the gonopore configuration (Figure 5). It is interesting to note that two lobsters with the female phenotype possessed male genetics, as identified with the *Po-iDMY* marker (marked in red, Figure 5C). To confirm the results, we collected those samples as another independent test for PCR with a *Po-iDMY* marker, the results of which were the same both times. This work indicates that *Po-IAG* silencing may affect sexual development, potentially leading to a change in sexual differentiation in TRL.

## 3. Discussion

In the context of aquaculture, an understanding of the regulation of sexual development pathways is essential for establishing a successful population in captivity and developing sex manipulation techniques. In the case of a new aquaculture species like TRL, it is necessary to attain a good knowledge of the molecular basis of the sexual development pathway with the objectives of characterizing key factors that can be manipulated to enable sex change and potentially increase growth rates. Here, we developed an assembly of the draft genome of a male TRL to aid in understanding key factors associated with sexual development in this species. This study found the genome size of the TRL male to be approximately 2.446 Gbp, which closely aligned with the estimated value of 2.46 Gbp (GenomeScope results). In comparison, our genome size is similar to the previous TRL male genome, which was reported to be 2.65 Gbp, and is larger than that of the TRL female genome, reported to be 1.93 Gbp [99,100]. Furthermore, the percentage of complete BUSCO and the percentage of fragmented signature homologs found in the female genome were 77.6%, and 15.5%, respectively, while they were 87% and 8.8% in the male genome [99]. Overall, our assembly of the male genome is more complete than that of the female genome, with increased annotated sequences, including the male-specific *Po-iDMY*. According to a summary of 37 crustacean genomes, the average number of protein-coding genes was approximately 25,000 genes per genome [110], ranging between 12,598 predicted genes in *Homarus americanus* [111] up to a striking 99,127 predicted genes in the female TRL genome [99]. Recently, a chromosome-level genome for the male *P.ornatus* identified 22,752 protein-coding genes, of which 99.20% were functionally annotated. Unexpectedly, about 129,814 protein-coding genes were annotated in our current assembly of the male TRL genome, much higher than the results of the previous studies. Further research is suggested to gain an accurate estimate of gene prediction, including improving genome assembly quality and optimizing gene prediction methods [110]. Interestingly, we were able to identify different genes to varying levels of coverage in the two male genomes, and thus, there is intrinsic value in producing multiple genomes for individual species, as comparative analysis will improve the reliability of the outcomes. Furthermore, the use of transcriptomic data from multiple tissues in the present genome assembly (nineteen tissues compared with eight tissues of the published male genome) is of importance, as showcased by the identification of *Po-iDMY*.

Importantly, we identified the complete sequence of the *Po-iDMY* gene and validated its use as a genetic sex marker, a critical component in the development of mono-sex populations in the future. In addition, to the best of our knowledge, *Po-iDMY*, a male-specific heterogametic (Y-linked) paralogue of the autosomal *Po-iDmrt1*, is the second sex-linked *iDmrt* gene to be identified in invertebrates. In the closely related ERL, *Sv-iDMY* was found to be a sex regulator gene, and it is surprising that iDMY does not appear to have the same function in TRL. In the case of ERL, *Sv-iDmrt1* expression increased rapidly during the first ten days of embryo development, then remained at relatively low levels throughout embryogenesis, although its expression increased slightly in later development of the embryos. Meanwhile, *Sv-iDMY* was lowly expressed during early embryo development, with expression dramatically increasing from D0 to D65. The embryonic stages during which the expression of *Sv-iDMY* was higher than that of *Sv-iDmrt1* were defined as the putative male sex-determining period (grey box in Figure 6). Taken together, Chandler et al. (2017) confirmed that Sv-iDMY is a master sex regulator in ERL [62] (Figure 6). In contrast, we showed here that the expression of *Po-iDMY* and *Po-iDmrt1* was very low for both genes throughout embryonic development in addition to during the metamorphic stages and in multiple adult tissues. These findings suggested that *Po-iDMY* may not tightly regulate IAG, as indicated in ERL.

Our bioinformatic predictions conducted structural analysis and revealed that the protein sequence of Po-iDMY was similar to that of its autosomal Po-iDmrt1, since they contain two DM domains and a TAD (Figure 7B). In addition, our observations on gene architecture have shed light on differences between the *Dmrt* genes of the two lobster species, potentially resulting in mechanistic variability in the functions of these proteins. The *Po-iDMY* peptide sequence is truncated, but contains a TAD at the C’ end, suggesting it does not suppress the transcriptional activity initiated by *Po-iDmrt1* (Figure 7B), which is seen in ERL. Indeed, the truncation of *Sv-iDMY* and lack of TADs negatively suppresses the transcriptional activity of the autosomal *Sv-iDmrt1*, which contains two TADs [74,112] (Figure 7A). Having said that, our finding showed that the *Sv-iDMY* and *Po-iDMY* structures are also similar, since they contain the male-specific sequence and can serve as genetic sex markers, which can facilitate the development of mono-sex production techniques. Taken together, *Po-iDMY* is simply a Y-linked *iDmrt1* paralogue that could serve as a male-specific marker in TRL. In addition, this finding indicated that the chromosomal mechanism of sex determination of *P. ornatus* is as a heterogametic sex (XY) similarly to ERL [69,90,113]. Using this information, we have achieved two essential elements that need to be considered for sexual manipulation in this species: (1) genetic sex markers; and (2) the chromosomal mechanism.

A lack of *Dmrt* expression during the period of sex determination during embryogenesis implied that other factors may be involved in this process in TRL, leading to *IAG* being implicated as a potential gene to manipulate for inducing sex change. However, our findings confirmed that *Po-IAG* is expressed in both males and females, suggesting that *Po-IAG* has additional functions in TRL. This is in line with the red swamp crayfish, *Procambarus clarkia* [86], and the mud crab, *Scylla paramamosain* [82], where *IAG* is widely expressed in multiple tissues of both males and females. In the successful case of *M. rosenbergii*, Ventura et al. (2012) [94] found that *IAG* expression was detected exclusively in the androgenic gland and started approximately 20 days after metamorphosis, prior to the appearance of external sexual characteristics. This revealed the perfect condition and time for silencing *IAG* to achieve a full functional sex reversal of males into neo-females in the giant freshwater prawn, *M. rosenbergii* [94,114]. According to our studies, unlike ERL, where individuals are sexually dimorphic upon metamorphosis, the external sexual characteristics of TRL are not evident prior to the fourth juvenile stage (J4), suggesting sexual plasticity up to this stage in TRL.

In addition, an interesting finding is that *Po-IAG* is expressed in early juvenile stages (J1), which leads to more evidence for the hypothesis that *Po-IAG* can be manipulated prior to J4 to induce sex change in TRL, and successful sex change can be validated by using *Po-iDMY* as a genetic marker. In our findings, after 2 months of dsRNA injection targeting *IAG*, based on the gPCR results of *Po-iDMY*, we found two individuals with a female phenotype which were potentially genetically males. Considering all aspects investigated here, it seems highly probable that silencing *Po-IAG* may be successful in sexual manipulation in TRL. However, further investigation and experimentation should continue until these animals mature to confirm the expected results.

## 4. Materials and Methods

### 4.1. Male Genome

#### 4.1.1. Sample Collection

For genomic DNA sequencing, a mature male TRL (*Panulirus ornatus*) was acquired from wild-caught stocks captured in the Torres Straits in 2019 and reared at the University of the Sunshine Coast (UniSC) for at least one month prior to dissection for tissue isolation, which was carried out after placing the animal in saltwater ice slurry until immobilized. The tail muscle was snap frozen in liquid nitrogen and kept at −80 °C for further work. Transcriptomic data across multiple tissues are also available for the same individual.

#### 4.1.2. High-Molecular-Weight Genomic DNA Extraction

Genomic DNA (gDNA) of the tail muscle tissue was extracted using salting-out followed by ethanol precipitation [115]. DNA from 10 mg of tail muscle was extracted using the following protocol. Tissue was ground with a pestle and mortar in liquid nitrogen and placed in a 1.5 mL tube supplemented with 300 µL of Cell Lysis Solution (10 mM Tris-HCl, pH 8.0, 25 mM EDTA, 0.5% SDS) and 30 µg of Proteinase K incubated at 55 °C for 3–16 h followed by RNase A treatment (3 µL, 4 mg/mL) at 37 °C for 1 h. Proteins and cellular debris were then removed by adding 100 µL of Protein Precipitation Solution (5 M Ammonium Acetate) and kept at room temperature for 10 min, followed by vigorous vortexing at high speed for 30 s to pellet proteins. Centrifugation was then conducted at 2500× *g* for 10 min, and the supernatant was collected and transferred into a fresh 1.5 mL tube containing 300 µL of isopropanol, which was inverted 25–50 times to precipitate genomic DNA followed by centrifugation at 6000× *g* for 5 min to collect the pellet. In the following step, 300 µL of 70% ethanol was used to wash the DNA pellet twice. The supernatant was discarded, and the pellet was air-dried for 10 min, then resuspended in 30 µL Tris-EDTA (ethylenediaminetetraacetic acid) and left overnight at room temperature. The integrity of the DNA sample was checked on a 0.8% agarose gel, and the sample was stored at 4 °C.

#### 4.1.3. Next-Generation Sequencing

Chromium 10× Genomic libraries construction and DNA sequencing (using Illumina NovaSeq 6000 (Illumina, Inc., San Diego, CA, USA), 150 base pair (bp), paired-end sequencing) was carried out by the Australian Genome Research Facility Ltd. (AGRF Ltd., Gehrmann Laboratories, Brisbane, QLD, Australia). The sequence read files of 4 cells were stored as FASTQ files. PacBio SMRTbell libraries were made using the PacBio template Prep Kit 1.0 and sequenced across 9 SMRT Cells 1 M on the Sequel System. High-Molecular-Weight (HMW) Genomic DNA (gDNA) was sheared using 26 G Blunt-End Needles (SAI Infusion-B26-150) to reduce its size and checked for shearing size with pulsed-field electrophoresis to ensure 10–20 kb insert sizes. Low output from the first 3 SMRT Cells led to whole-genome amplification being carried out to remove possible contaminants prohibiting sequencing. This was carried out with a Qiagen REPLI-g Mini Kit (150023) (Qiagen Sciences, Germantown, MD, USA), followed by branch removal with T7 Endonuclease I prior to SMRTbell library prep. This method enabled better outputs from the SMRT cells for improved genome assemblies. Pacific BioSciences library preparation and sequencing of 9 SMRT cells of Sequel I was performed at the Queensland University of Technology (QUT).

#### 4.1.4. Genome Assembly and Annotation

The 10× linked-read files (PRJNA952321), consisting of two pairs of read files and PacBio data (PRJNA952321), have a five times bigger data volume than the 10× decontaminated using Kraken version 2.0.8b [116] with their default database. After decontamination, the 10× linked-read files, PacBio data, combined 10× linked-read files, and PacBio data were analyzed and de novo assembled using Supernova version 2.1.1 [117]. The draft genome was mapped using subreads of the PacBio data with minimap2. Scaffolding was then performed using LRScaf (version 1.1.12). The output of LRScaf was then mapped using the male transcriptome assembly downloaded NCBI Bioproject PRJNA903480 and scaffolded using L_RNA_scaffolder (https://github.com/CAFS-bioinformatics/L_RNA_scaffolder; https://github.com/CAFS-bioinformatics/L_RNA_scaffolder/blob/src/README.md (accessed on 6 July 2023)). Both raw reads from the 10 X linked-read and PacBio data were used to fill N-gaps in the final genome assembly using TGS-GapCloser (https://github.com/BGI-Qingdao/TGS-GapCloser (accessed on 5 August 2023)). The size, heterozygosity, and repetitive sequences of the genome were estimated by GenomeScope 2.0 (https://github.com/tbenavi1/genomescope2.0 (accessed on 6 September 2023)) [106]. Finally, genome completeness was assessed with Benchmarking Universal Single-Copy Orthologs (BUSCO) version 5.4.2 [118] using arthropoda_odb10 and metazoa_odb10. Transposable elements (TEs) were annotated as previously described [108], using the automated Earl Grey TE annotation pipeline (Appendix A) (https://github.com/TobyBaril/EarlGrey (accessed on 17 October 2023)). BRAKER predicted gene models [119,120] with the evidence from multi-tissue (eyestalks; brain; thoracic ganglia; antennal gland; testis; sperm duct (proximal, medial, distal); hepatopancreas; gill; heart; midgut; hindgut; tail muscle; epidermis; fat tissue; and hemocytes) transcriptomes of Ventura et al. (2020) [78] and Arthropoda proteins from OrthoDB11.

### 4.2. Identification of the Full Sequence of the Male Y-Linked iDmrt Paralogue, Po-iDMY

#### 4.2.1. Sample Collection

Two mature TRLs, one male and one female, were acquired from wild-caught stocks captured in the Torres Straits in 2019 and reared at UniSC. RNA was extracted from their walking legs using an RNAZolRT protocol [121]

#### 4.2.2. Identify the Complete Sequence of Po-iDMY

The RNA samples then served as a template for generating the 3′ and 5′ rapid amplification of *Po-iDMY* cDNA using a Clontech SMARTer™ RACE kit (BD Biosciences, Franklin Lakes, NJ, USA), according to the manufacturer’s instructions [122]. First, primary PCR was performed with the gene-specific forward primer *Po-iDMY*-GSP1_F (designed based on the similarity of the *Sv-iDMY* sequence to other decapods) and a universal primer mix (UPM) from the RACE kit as a reverse primer. PCR conditions were as described in Ventura et al. (2015) [123]. Then, *Po-iDMY*-NGSP1_F was used as a forward primer and a universal primer mix (UPM) from the RACE kit as a reverse primer for the secondary PCR (nested PCR). The DNA templates were used by diluting 5 µL of the primary PCR product into 245 µL of Tricine-EDTA buffer. Nested PCR was carried out using the manufacturer’s protocol. The PCR products were cloned and Sanger-sequenced at the Australian Genome Research Facility Ltd. (AGRF Ltd., Gehrmann Laboratories, Brisbane, QLD, Australia).

#### 4.2.3. Sequence Analysis

The molecular weight and isoelectric point of *Po-iDMY* proteins were predicted by ExPASY’s analysis tool version 3.0 (https://www.expasy.org/resources/compute-pi-mw (accessed on 2 March 2024)). Clustal Omega version 1.2.34 was used to align the sequence of our transcripts (https://www.ebi.ac.uk/jdispatcher/msa/clustalo (accessed on 3 March 2024)) with the partial *iDMY* sequence found by Ventura et al. (2020) [78] and domain architecture defined with NCBI Conserved Domains v3.20 (https://www.ncbi.nlm.nih.gov/Structure/cdd/wrpsb.cgi (accessed on 10 March 2024)). Transactivation domains (TADs) were investigated by the Nine Amino Acids Transactivation Domain (9aaTAD) Prediction Tool [124,125,126] with the “less stringent pattern” and a “100% match” criteria. ATGpr [127] (https://atgpr.dbcls.jp/ (accessed on 23 April 2024)) was used to predict the initiation codons and stop codons in the cDNA sequences.

#### 4.2.4. Validation of Male-Specific Po-iDMY Marker

TRLs were reared from egg at the Institute for Marine and Antarctic Studies (IMAS) aquaculture facility in Taroona, Tasmania. Two batches of swimming leg samples, from 20 mature males and 20 mature females, stored in 70% ethanol, were isolated at IMAS and transported to UniSC. According to the manufacturer’s instructions, genomic DNA was extracted using a REDExtract-N-Amp Tissue PCR Kit (Sigma, St Louis, MI, USA). The IMAS team validated the sex of each sample, and this was not revealed to the UniSC team until after the PCR results were disclosed (a blind test). The Primer3 software [128] was used to design primers for the sex-specific *Po-iDMY* marker, Po-iDMY R, and Po-iDMY F (Table 1). PCR conditions included preheating to 94 °C for 3 min, followed by 35 cycles of 94 °C for 30 s, 60 °C for 30 s, and 72 °C for 2 min, followed by final elongation at 70 °C for 10 min and a final hold at 4 °C. As a positive control, the *Po-Dmrt* gene was amplified for each sample tested using primers named Po-Dmrt F and Po-Dmrt R (Table 1) with the following PCR conditions: 94 °C for 3 min, followed by 30 cycles of 94 °C for 30 s, 58 °C for 30 s, and 72 °C for 2 min and final elongation at 70 °C for 2 min. PCR products were electrophoresed on a 2% agarose gel and visualized under UV light with ethidium bromide.

### 4.3. iDMY Ontogeny

A BLAST search (version 2.16.0+) using *Po-iDMY* sequences was carried out against the transcriptome database of embryonic development [102], metamorphic stages [129], and adult tissues [78] through crustybase.org [109]. The Fragments Per Kilobase of transcript per Million mapped reads (FPKM) values of *Po-iDMY* transcripts were then investigated using these three databases.

### 4.4. Correlation Between Po-IAG Expression and the Appearance of Sexual Characteristics

Twelve hatchery-reared TRLs at the first molt following metamorphosis to juvenile (J1) were sampled at IMAS, snap-frozen in liquid nitrogen, and kept at −80 °C until they were shipped to UniSC on dry ice, where RNA from the fifth walking leg was extracted as described above. A total of 1 µg of RNA was then reverse-transcribed by using a Tetro cDNA Synthesis Kit (Bioline, London, UK) following the manufacturer’s instructions, and the cDNA was used for RT-PCR, using *Po-IAG*-specific primers, with primers for insulin-like growth factor binding protein (IGFBP), which is expressed across tissues in abundance [78], as a positive control (Table 1). Since *Po-IAG* expression was very low and no amplicons were detected following one round of PCR (with *Po-IAG*/394nt primers), a nested PCR with another set of primers (*Po-IAG*/189nt) was performed.

In addition, to identify the gender of these juveniles when their external characteristics had not yet developed, their genomic DNA was amplified by using the male-specific *Po-iDMY* primers (Table 1, as used in Section 2.4). Genomic DNA was extracted from the tail muscle using a REDExtract-N-Amp Tissue PCR Kit (Sigma, St Louis, MI, USA). The PCR conditions were as follows: 94 °C for 3 min, followed by 35 cycles of 94 °C for 30 s, 60 °C for 30 s, and 72 °C for 2 min and then final elongation at 70 °C for 10 min and a final hold at 4 °C. PCR products were electrophoresed on a 2% agarose gel and visualized under UV light with ethidium bromide.

**Table 1 ijms-26-05149-t001:** Primers used in this study.

Primer Name	Gene/Amplicon Size (nt)	Primer Sequence 5′–3′	Template
Po-iDMY-GSP1_F	*iDMY*	GGACACCAAGCTACAGAAGTGCGAC	cDNA
Po-iDMY-NGSP1_F	*iDMY*	GGCGTTATGAAAGAGAAGCGGGCCC	cDNA
Po-IAG F	*Po-IAG/394nt*	TCTCCTCCTACAACGTGACG	cDNA
Po-IAG R		TGTCGTAGCTCAGTGTCACT	
Po-IAG F	*Po-IAG/189nt*	CAAGTCTTACATCGGCAGCC	cDNA
Po-IAG R		TTCGTGATAGGAGGGTTGCC	
Po-IGFBP F	*Po-IGFBP/309nt*	GGAGGGATCTTCGTTGTTGC	cDNA
Po-IGFBP R		TGACCCACATACAGGATCCG	
Po-iDMY F	*Po-iDMY/208nt*	AGGTTGGGAAGTACCCAGTG	gDNA
Po-iDMY R		GTCGCACCTCTCAAAGAACC	
Po-Dmrt R	*Po-Dmrt/600nt*	GCAGCCTGAATATGAGGGGT	gDNA
Po-Dmrt F		AGTAAGGCAAGTTGACGGGA	

### 4.5. In Vivo Panulirus Ornatus IAG Silencing

Sixty TRLs (0.1192 g ± 0.0035) were reared at IMAS from the J1 stage for experiments. Juvenile lobsters were cultured in plastic tanks (27.5 × 41.5 × 24 cm) at 28 °C, with a 12:12 L:D photoperiod. IMAS commercial-in-confidence feed was supplied into four feeding rations over 24 h. Tanks were siphoned daily to remove uneaten food and waste. Sixty J1 lobsters were split into four tanks (15 lobsters per tank) and injected through the sinus at the base of the 5th walking leg with 2 µL (5 µg/µL) of *Po-IAG* dsRNA using a microapplicator (Burkard, UK) combined with an insulin syringe. The dsRNA was purchased from Genolution (https://agrorna.genolution.co.kr/, accessed on 23 September 2023). The injection was repeated weekly for 8 weeks until the lobsters reached juvenile stage 4 (J4).

After the injection period, phenotypic sex confirmation was conducted using photos of gonopores, and genetic sex was confirmed using the genetic sex marker (*Po-iDMY*) at juvenile stage J6, when the external characteristics were easy to identify. To identify the genetic sex of the lobsters, pleopods were collected and kept in 80% ethanol at 4 °C. Genomic DNA was extracted from the pleopods as described previously (Section 2.4) and used to determine whether the animals were genetically male or female by identification of the *Po-iDMY* gene in males.

## Figures and Tables

**Figure 1 ijms-26-05149-f001:**
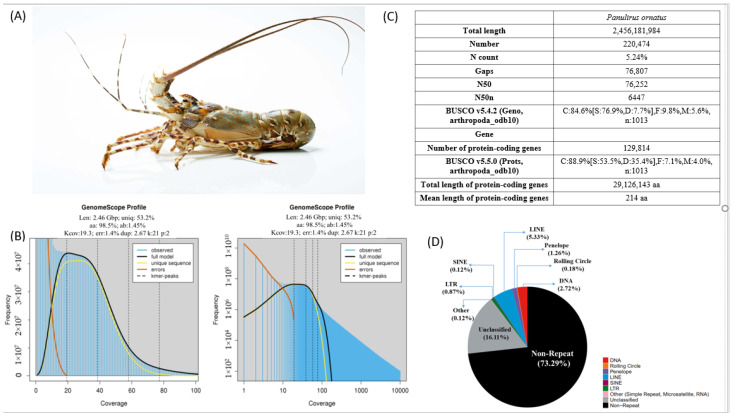
(**A**) Panulirus ornatus adult male (2.5 kg); (**B**) Genomescope report—coverage of the kmer (x) by kmer counts (y) with k-mer = 21 with Len, estimated total genome length; Uniq, unique portion of the genome (not repetitive); Het, heterozygosity rate; Kcov, K-mer coverage for the heterozygous bases; Err, error rate; Dup, duplication rate, where the blue bar in the graph represents the observed K-mer, and the yellow and orange lines in the graph represent the errors and unique sequences, respectively—on the left, the coverage represented is the linear plot, and on the right, the coverage represented is log-scale; (**C**) statistics of the genome assembly generated in this study; (**D**) repetitive elements distribution.

**Figure 2 ijms-26-05149-f002:**
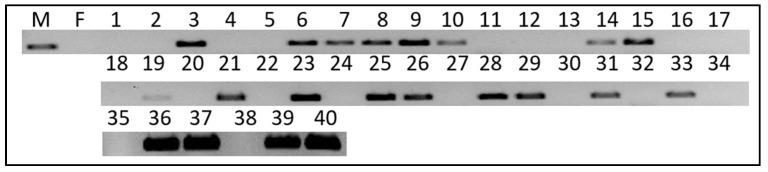
Validation of the male-specific *Po-iDMY* marker by PCR of genomic DNA. Forty blind TRL samples were used to validate the male-specific *iDMY*. The male genotype is defined as the presence of a genomic sex marker where no specific marker was amplified in the females. According to these results, 21 samples were indicated as males and 19 samples were females; thus, 1 sample failed, and the success rate was nearly 100%. M—males; F—females; number (1,2…40)—sample order.

**Figure 3 ijms-26-05149-f003:**
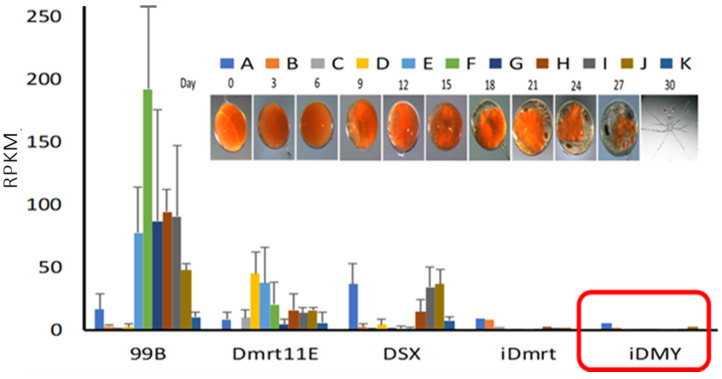
Expression of key sexual development transcripts across embryonic development of the ornate spiny lobster *Panulirus ornatus*. The expression of transcripts encoding the transcription factors coding for a DM domain (Dmrt family genes) was calculated as reads per kilobase per million reads (RPKM) in RNA-Seq libraries of multiple TRL embryonic stages (A–K representing days 0 to 30, sampled every 3 days, with triplicates per stage). While several Dmrt family genes were expressed in embryos (99B, Dmrt11E, DSX), none were male-specific. The male-specific *Po-iDMY* showed negligible expression across all examined embryonic stages (marked in red).

**Figure 4 ijms-26-05149-f004:**
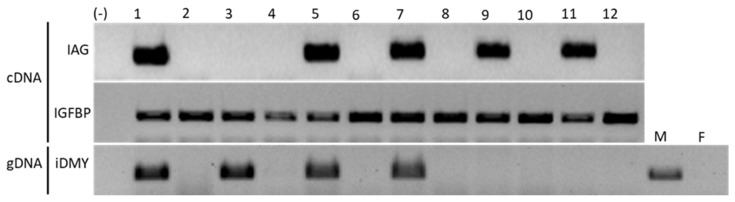
RT-PCR of RNA and PCR of genomic DNA extracted from 12 TRL juveniles (J1). RT-PCR was applied for detection of *IAG* expression by using *IAG*-specific primers, with *IGFBP* (insulin-like growth factor-binding peptide) as the positive control. The *iDMY* marker was used to identify the sex of the samples by genomic DNA-based PCR. Juveniles were inferred to be females based on the absence of *iDMY*. There was no correlation between the presence of the male-specific marker and *IAG* expression, as evidenced by individual 3 (genetic male with no *IAG* expression) and individuals 9 and 11 (genetic females with *IAG* expression). M—males; F—females.

**Figure 5 ijms-26-05149-f005:**
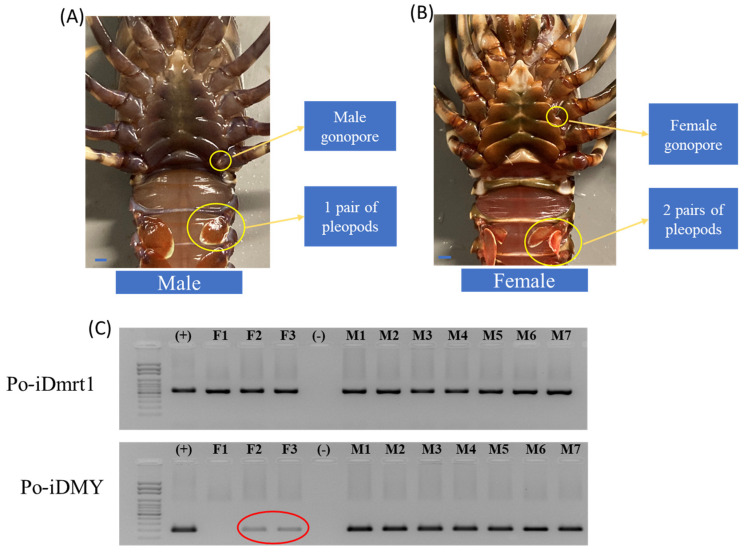
PCR of genomic DNA extracted from 10 *Panulirus ornatus* adults that were injected with dsRNA weekly to silence the *IAG* gene for 2 months (from J1 to J4). Phenotypic sex was determined based on the development of gonopores based on the third walking leg in males and the third walking leg in females. (**A**) Male lobsters with gonopores on the fifth walking legs and one pair of pleopods; (**B**) female lobsters (potentially male genetic) with gonopores on the third walking leg and twp pairs of pleopods; (**C**) the *Po*-*iDMY* marker was used to identify the genetic sex of the samples by genomic-based PCR. Samples were inferred to be genetic females based on the absence of this marker. *Po-iDmrt1* was used as a positive control for DNA integrity. Two individuals that were phenotypic females showed the male-specific *Po-iDMY* marker (marked in red). This test was repeated twice, independently (see the results for the second time in the Appendix A), with the same result. M—males; F—females; (+): positive control (male sample); (−): negative control (water). Scale bar: 1 mm.

**Figure 6 ijms-26-05149-f006:**
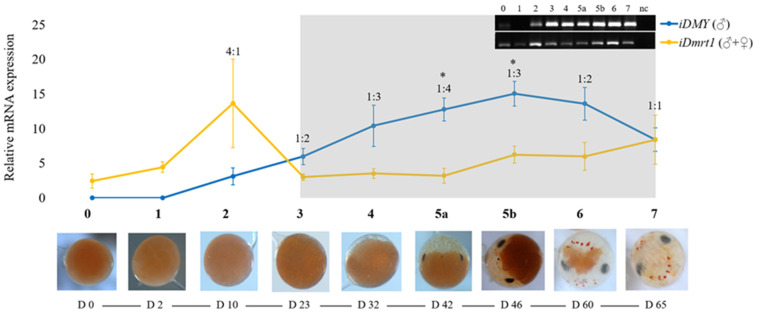
ERL *iDmrt1* and *iDMY* expression during embryogenesis (D), D0—day 0—to hatching on D65—day 65; the ratio of *iDmrt1*: *iDMY* expression is presented at each stage, and the putative male sex-determining period is in the grey box; stages at which *iDMY* > *iDmrt1* (* *p* < 0.01), supporting the PCR gel image in the top right (modified and represented with permission from Chandler et al. (2017) [62]).

**Figure 7 ijms-26-05149-f007:**
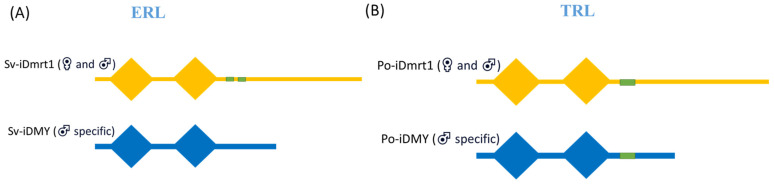
Comparing the domain architecture of predicted Dmrt proteins from ERL (Sv) (**A**) and TRL (Po) (**B**); DM domains are represented by diamonds, and the predicted transactivation domains (TADs) are shown in green rectangles.

## Data Availability

Data is contained within the article and Appendix A.

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
