# Peer review of "Roles of a Y-Linked iDmrt1 Paralogue and Insulin-like Androgenic Gland Hormone in Sexual Development in the Tropical Rock Lobster, Panulirus ornatus"

_ijms, 2025, doi:10.3390/ijms26115149_

Round 1

Reviewer 1 Report

Comments and Suggestions for Authors

This study offers some novel insights into sexual development mechanisms in the tropical rock lobster (Panulirus ornatus). However, several critical limitations in experimental design and data interpretation currently undermine the robustness of its conclusions. Substantial revisions and additional validation are required to strengthen the claims. As outlined below:

​​1. The validation of the genome assembly and annotation requires strengthening, since the reported 129,814 protein-coding genes significantly exceed previous estimates (e.g., 22,752 genes in the chromosome-level male genome by Ren et al., 2024).

​​2. The conclusion that Po-iDMY as the sex markers lacks robustness, wild populations or different developmental stages remains untested. Additionally, a positive control was needed for the female samples.

​​3. The IAG RNAi experiment lacks negative controls, which undermining the conclusion about sex reversal.

​​4. ​​L42​​: Add "etc." after "spiny lobsters"; ​​provide scale bars for Fig. 5A-B.

Author Response

Thank you for taking the time to review this manuscript and for your helpful comments. Your advice is greatly appreciated. We have amended the manuscript following the changes to the prose that you suggested on the marked manuscript. Please find the detailed responses below and the revisions in the re-submitted files.

Comments 1: The validation of the genome assembly and annotation requires strengthening, since the reported 129,814 protein-coding genes significantly exceed previous estimates (e.g., 22,752 genes in the chromosome-level male genome by Ren et al., 2024).

Response 1: Thank you for pointing this out. We agree with this comment and wrote in our paper at line 505-509: “Unexpectedly, about 129,814 protein-coding genes were annotated in our current assembly of the male TRL genome, much higher than the results of the previous studies”. Whilst Ren et al. (2024) produced a chromosome-level male genome assembly, iDMY was not identifiable in it, yet it was found in our assembly, pointing to the added benefit of our detailed draft assembly in complementing the existing data.

Please note that the key highlight of the current study is not the genome assembly but rather the development of a monosex biotechnology based on the identification and characterization of iDMY as a genetic sex marker and IAG as the masculinization hormone.

Comment 2: The conclusion that Po-iDMY as the sex markers lacks robustness, wild populations or different developmental stages remains untested. Additionally, a positive control was needed for the female samples

Response 2: We do not agree that different developmental stages will be affected when using the sex markers. The reason is that we worked on genomic DNA (gDNA). gDNA does not change during those stages. In addition, our samples were bred from a wild population where this marker is consistently validated. For the purpose of establishing a viable monosex biotechnology, this marker validation within this population is sufficient. Additional validation in other wild populations is out of scope for the current study and can be done in future by other research teams if they wish to establish a similar technology.

The method used includes inherent positive controls. We obtained blinded samples, which means that the person at UniSC who ran the PCRs did not know which were males and which were females, but the staff at IMAS who provided the samples knew accurately which are females by examining their gonopores before collecting the tissue samples. The validation following PCR was nearly 100% correct, with only one female inaccurately shown to be a male using our sex marker. Our sex marker proved to be robust in this test, where all known females can be regarded as positive controls.

Comment 3: The IAG RNAi experiment lacks negative controls, which undermining the conclusion about sex reversal.

Response 3: Unlike Macrobrachium rosenbergii where numerous juveniles can be reliably sourced and cultured, this study aims at developing the monosex biotechnology for the ornate spiny lobster, to a proof of concept level. Note that the access to juveniles is scarce and the ability to maintain them under culture conditions and establish robust gene silencing protocols is highly challenging. The experiment clearly identifies IAG silenced individuals which are determined to be males using the genetic markers and display a female phenotype. The ultimate proof would be breeding these animals and showing change in sex ratios, as was done in M. rosenbergii. However, unlike M. rosenbergii, this would take at least 7 years in the ornate lobster.

Comment 4: ​​L42​​: Add "etc." after "spiny lobsters"; ​​provide scale bars for Fig. 5A-B.

Response 4: corrected as advised.

Reviewer 2 Report

Comments and Suggestions for Authors

In the present study, authors aimed to investigate the roles of a Y-linked iDmrt1 paralogue and Insulin-Like Androgenic Gland Hormone in sexual development in the Tropical Rock Lobster. The sexual marker was identified in Po-iDMY, and IAG was identified to cause the sexual reverse in Tropical Rock Lobster. Generally, the present study is interesting. However, some revision should be investigated.

In the abstract, authors should briefly introduce that why they aimed to study the mechanism of sexual development in the Tropical Rock Lobster.

In abstract, authors should clearly describe their results and conclusion. The results descripted are too ambiguous.

Line 28-30, why Po-iDMY did not tightly regulate Po-IAG in this species, when you concluded that silencing Po-IAG could potentially induce a sex change.

In the last sentence of Introduction, please clearly describe what experiments authors aimed to perform, and which technique authors aimed to use.

Line 165-166, how did you prepare the samples? How many tissues were pooled? Are there any biological replicates?

Authors should measure the expressions of Po-iDMY by qPCR. Not only calculated through the data generated by RNA-Seq.

N50 of 76 kbp is too low.

3.1.5, provided the sequence of Po-iDMY, and labelled the conserved domains. In addition, please perform the phylogenetic tree analysis of Po-iDMY.

Line 530, why Po-iDMY and Po-iDmrt1 were not significantly expressed throughout embryonic development, during the metamorphic stages, and in multiple adult tissues, and then authors conclude that Po-iDMY may not tightly regulate IAG.

Author Response

Thank you for taking the time to review this manuscript and for your helpful comments. Your advice is greatly appreciated. We have amended the manuscript following the changes to the prose that you suggested on the marked manuscript. 

Please find the detailed responses below and the revisions in the re-submitted files.

Comment 1: In the abstract, authors should briefly introduce that why they aimed to study the mechanism of sexual development in the Tropical Rock Lobster.

Response 1: Corrected as advised.

Comment 2: In abstract, authors should clearly describe their results and conclusion. The results descripted are too ambiguous.

Response 2: corrected as advised.

Comment 3: Line 28-30, why Po-iDMY did not tightly regulate Po-IAG in this species, when you concluded that silencing Po-IAG could potentially induce a sex change.

Response 3: The insulin-like androgenic gland hormone (IAG) is a key downstream mediator governing sexual differentiation. If IAG is not regulated by a master sex regulator, it can control sexual differentiation by itself. This is unlike the eastern spiny lobster Sagmariasus verreauxi, where we clearly showed that iDMY is expressed in males at early egg development stages. In the ornate lobster (this study) iDMY is conserved in sequence and position on the Y chromosome but it is not expressed during embryonal development, As a result, when we turn it off by silencing, we potentially induce a sex change. 

Comment 4: In the last sentence of the Introduction, please clearly describe what experiments the authors aimed to perform, and which technique the authors aimed to use.

Response 4: In the final paragraph of the Introduction we have described the experiments and techniques used (line 159 to 173).

Comment 5: Line 165-166, how did you prepare the samples? How many tissues were pooled? Are there any biological replicates?

Response 5: Sample preparation for the Transcriptomic data across multiple tissues was described in the paper published by our group as cited (Ventura et al. 2020b):

Ventura T., Chandler J.C., Nguyen T.V., Hyde C.J., Elizur A., Fitzgibbon Q.P. & Smith G.G. (2020b) Multi-Tissue transcriptome analysis identifies key sexual development-related genes of the ornate spiny lobster (Panulirus ornatus). Genes 11, 1150.

Comment 6: Authors should measure the expressions of Po-iDMY by qPCR. Not only calculated through the data generated by RNA-Seq.

Response 6: Po-iDMY expression is very low, therefore qPCR would be a futile effort in an attempt to validate the low, nearly non-existent expression of this gene. Given that the highlight of the manuscript is establishing the proof of concept of a monosex biotechnology, the validation of iDMY as a genetic sex marker addresses the research question.

Comment 7: N50 of 76 kbp is too low.

Response 7: Agreed! Yet, the added value of this draft assembly is in identifying genes that were not identified in a previous assembly.

Please note that the key highlight of the current study is not the genome assembly but rather the development of a monosex biotechnology based on the identification and characterization of iDMY as a genetic sex marker and IAG as the masculinization hormone.

Comment 8: 3.1.5, provided the sequence of Po-iDMY, and labelled the conserved domains. In addition, please perform the phylogenetic tree analysis of Po-iDMY.

Response 8:

- The sequence of Po-iDMY protein was provided in the Supplementary Data S4

- The labelled conserved domains of Po-iDMY protein from male and female were provided in the Figure 7. B

- Po-iDMY is a male-specific heterogametic (Y-linked) paralogue of the autosomal Po-iDmrt1 found in TRL, and just a second sex-linked iDmrt gene identified in invertebrates. The first iDMY was found in Sagmariasus verreauxi by our group (see reference below). Therefore, we could not perform the phylogenetic tree analysis.

Chandler J.C., Fitzgibbon Q.P., Smith G., Elizur A. & Ventura T. (2017) Y-linked iDmrt1 paralogue (iDMY) in the Eastern spiny lobster, Sagmariasus verreauxi: The first invertebrate sex-linked Dmrt. Developmental biology 430, 337-45

Comment 9: Line 530, why Po-iDMY and Po-iDmrt1 were not significantly expressed throughout embryonic development, during the metamorphic stages, and in multiple adult tissues, and then authors conclude that Po-iDMY may not tightly regulate IAG.

Response 9:

- iDMY is a male-specific heterogametic (Y-linked) paralogue of the autosomal iDmrt1, and iDmrt1 emerges as a pivotal upstream factor in the integrated sexual development cascade. (Line 86-92)

- The insulin-like androgenic gland hormone (IAG) is another crucial downstream mediator governing sexual differentiation , affecting the appearance of primary and secondary male sexual characteristics (line 128-130)

 - In our previous study, Sv-iDMY was found to be expressed at an early stage in embryonic development that dominantly suppressed its autosomal paralogue (iDmrt1), indicating that Sv-iDMY is a master sex regulator in ERL and plays a role as the master sexual development regulator upstream of IAG (from line 93-99). However, Po-iDMY and Po-iDmrt1 were not significantly expressed throughout embryonic development, during the metamorphic stages, and in multiple adult tissues. Our findings indicated no evidence supporting the notion that the male-specific gene Po-iDMY has a master sex-determining function in this species (439-441). Therefore, it is unlikely that Po-iDMY would regulate IAG, which is such an important hormone involved with sexual development in crustaceans.

Reviewer 3 Report

Comments and Suggestions for Authors

I found this paper, quite interesting as being able to find a sex specific gene that is only differentially expressed during an early developmental stage, as the author stated allows them to potentially regulate a sex early on for culturing crabs or rock lobsters in that could be a big benefit for agriculture to keep down predation as they mentioned.

 The 10 X analysis looks fine with the description for this gene. I think it’s really important for other researchers to maybe look at other species and see if this is a common gene that others might be able to target for other species of crustaceans.

The details of the paper are fine. All the methods are well described …. analysis genome sequencing assembly. I think others will benefit from this report and that they can use this as a foundation to further investigations. If one can actually manipulate this gene expression that would be great.

 Injecting the double strand RNA was and finding genetic males shows promise . Trying this with shrimp  the outcome might be faster due to the quick developmental stages.

The manuscript is well written, and details are explained. The Discussion pulls the data together and illustrates what the next steps in future investigations.

I do think Figure 1 could be presents in a column format so each part could be enlarged. When printed as a PDF it was a bit small to read text.

Author Response

Crustacean production would benefit from manipulation of stocks to yield the faster-growing sex. This proves more complicated for decapod crustaceans than for finfishes, and monosex production of only Macrobrachium rosenbergii has been achieved. Progress for other species will come from improved understanding and effective manipulation of their respective sex determination systems.

You appreciate the challenges in developing a monosex biotechnology for a spiny lobster species, and we thank you for your positive support of this study. Thank you for taking the time to review this manuscript and for your helpful comments. Your advice is greatly appreciated.

One suggestion below has been addressed as follows: 

Comment: I do think Figure 1 could be presents in a column format so each part could be enlarged. When printed as a PDF it was a bit small to read text.

Response:

Thank you for your suggestion, when the figure is viewed online it will be easier to read.

Round 2

Reviewer 1 Report

Comments and Suggestions for Authors
  1. For the identification of genetic sex, two pairs of primers or multiplex PCR were usually used to avoid the potential DNA quality and PCR reaction problems.
  2. I understand the difficults for the lobsters' researches, I suggested the author explains this problems in the manuscript for the potential readers.

Author Response

Thank you for taking the time to review this manuscript again and for your helpful comments. Your advice is greatly appreciated. We have amended the manuscript following the changes to the prose that you suggested on the marked manuscript. 

Point-by-point response to Comments and Suggestions

Comment 1. For the identification of genetic sex, two pairs of primers or multiplex PCR were usually used to avoid the potential DNA quality and PCR reaction problems.

Response 1: Corrected as advised. Please look at lines 275-278, 413-417, Table 1, and Supplementary Data S7.

Comment 2. I understand the difficults for the lobsters' researches, I suggested the author explains this problems in the manuscript for the potential readers.

Response 2: Corrected as advised. Please look at lines 49-57

Reviewer 2 Report

Comments and Suggestions for Authors

I have no futher comments.

Author Response

No further comments made by the reviewer.